# Quantum Identity Authentication in the Counterfactual Quantum Key Distribution Protocol

**DOI:** 10.3390/e21050518

**Published:** 2019-05-23

**Authors:** Bin Liu, Zhifeng Gao, Di Xiao, Wei Huang, Zhiqing Zhang, Bingjie Xu

**Affiliations:** 1Postdoctoral Station of Computer Science and Technology, College of Computer Science, Chongqing University, Chongqing 400044, China; 2Science and Technology on Communication Security Laboratory, Institute of Southwestern Communication, Chengdu 610041, China; 3Chongqing University–Uninversity of Cincinnati Joint Co-op, Chongqing University, Chongqing 400044, China

**Keywords:** quantum identity authentication, quantum key distribution, counterfactual quantum communication

## Abstract

In this paper, a quantum identity authentication protocol is presented based on the counterfactual quantum key distribution system. Utilizing the proposed protocol, two participants can verify each other’s identity through the counterfactual quantum communication system. The security of the protocol is proved against individual attacks. Furthermore, according to the characteristics of the counterfactual quantum key distribution system, we propose an authenticated counterfactual quantum key distribution protocol based on a novel strategy of mixing the two types of quantum cryptographic protocols randomly. The authenticated quantum key distribution can also be used to update the extent of the authentication keys.

## 1. Introduction

Quantum mechanics has produced immense influence in information security. The widely used public key cryptography algorithms such as the RSA public key algorithm are facing serious threat of quantum computation [1]. Meanwhile, quantum computation also promotes new kinds of cryptographic protocols that can combat the powerful computation capability of quantum computer. Interestingly, quantum mechanics could be a sharp spear to break cryptographic systems and also a strong shield to protect our privacy. Research shows that quantum key distribution (QKD) can provide information theoretic security between two distant and authenticated parties [2,3,4,5]. Various QKD protocols have been proposed utilizing different quantum coding technologies, as well as the other types of quantum cryptographic protocol, such as quantum secure direct communication [6,7,8,9,10,11], quantum secret sharing [12,13,14,15,16,17], quantum private querying [18,19,20,21,22,23] and so on [24,25,26].

Counterfactual QKD protocols employ a very interesting coding method where the valid key bits are generated when no photons have been transmitted in the public channel. Since no photons to be intercepted and captured for the signals of the valid key bits, it is very difficult for the adversaries to carry on an effective attack. Because of the above characteristics, the counterfactual QKD has attracted a lot of attention since its first appearance. In 2009, Noh proposed a QKD protocol [27] inspired by the counterfactual phenomena in quantum world [28] and the counterfactual computation [29]. The next year, Sun et al. proposed a high efficiency version of counterfactual QKD utilizing more beam splitters [30]. The same year, Yin et al. proved the security of Noh’s protocol strictly [31]. During the next few years, many experiments on counterfactual QKD protocol have been performed [32,33,34,35]. In the theoretic study of counterfactual quantum communication technology, some scholars analyzed the security of counterfactual QKD on real environment [36,37,38,39,40,41], while others proposed other types of quantum cryptographic protocols with counterfactual quantum communication technology, such as direct quantum communication [42,43,44], quantum private query [45], and so on [46,47,48,49,50,51,52,53].

As described above, the counterfactual QKD protocol has been proven secure in theory [31], however, the security is based on several necessary conditions, such as the perfect quantum detectors, the perfect single-photon source, true random number generator and so on. Secure and reliable identity authentication is one of the key requirements of the security of the counterfactual QKD. Realizing identity authentication with quantum technology has many potential advantages such as higher security, higher efficiency and immunity to certain kinds of replay attacks. Therefore, we propose a quantum identity authentication (QIA) protocol that can be used in the counterfactual QKD protocol to identify the communication parties. Furthermore, due to the characteristics of the counterfactual quantum communication technology, the combined processes of the counterfactual QKD protocol and the proposed QIA protocol can be used to extend the length of authentication keys with almost arbitrary expanded proportion. This paper is organized as follows. In Section 2, we briefly review the processes of the counterfactual QKD and an alternative version, which our QIA protocol is based on. The specific process of the counterfactual QIA protocol is proposed in Section 3. With the QIA protocol in Section 3, we propose an authenticated counterfactual QKD protocol in Section 4. A brief conclusion is given in Section 5.

## 2. Review of the Counterfactual Quantum Key Distribution Protocol

The main purpose of this paper is to verify the participants’ identities in the counterfactual QKD system. As a foundation protocol, the counterfactual QKD protocol [27] is briefly introduced in this section. Utilizing the interference system in Figure 1, the counterfactual QKD can help Alice and Bob generate a secure key based on the signals where no photons have traveled through the public channel. Note that, in Figure 1, C is the optical circulator; OD is the optical delay to make the two paths *a* and *b* be the same; OL is the optical loop and SW is the optical switch, which help Bob choose the pulse in specific polarization to the detector D_2_; FM is the faraday mirror, which reflects the pulse while turns the state of the pulse to the orthogonal polarization; and D_1_ can discriminate the polarizations of the pulse. The processes can be described as follows.

(1) At the beginning, Alice triggers the single-photon source S to emit a short optical pulse containing a single photon at a certain point in time. The photon is prepared in either horizontal polarization |H〉, which represents the classical bit 0, or vertical polarization |V〉, which represents 1. Afterwards, the pulse will be divided into two paths, *a* and *b*, when it passes the beam splitter (BS). The whole system can be described as one of the two orthogonal states
(1)$$\begin{array}{c} \sqrt{T}{|0\rangle }_{a}{|V\rangle }_{b}+i\sqrt{R}{|V\rangle }_{a}{|0\rangle }_{b}, \end{array}$$
(2)$$\begin{array}{c} \sqrt{T}{|0\rangle }_{a}{|H\rangle }_{b}+i\sqrt{R}{|H\rangle }_{a}{|0\rangle }_{b}, \end{array}$$
where *R* and *T* are the reflectivity and transmissivity of BS, and *R*+*T* = 1. The state ${|0\rangle }_{k}$ represents the vacuum state in the path *k*, where k∈{a,b}.

(2) Bob randomly chooses a bit 0 or 1 and, utilizing the polarizing beam splitter (PBS) and the optical loop (OL), switches different polarized pulse to the detector $D_{2}$ according to the above bit. Precisely, if Bob chooses 0 (1), he switches the pulse in the state |H〉 (|V〉) to the detector $D_{2}$. In fact, when the pulse in path *b* reaches the PBS at Bob’s side, it would directly go to the optical switch (SW) if the pulse were horizontally polarized, and if the pulse were vertically polarized, it would be reflected by the PBS, pass through the OL, be reflected by PBS again, and then go to SW. Thus, if the pulse were in state |V〉, it would arrive at SW a certain period of time (L/c, where *L* is the length of OL and *c* is the speed of time) later than the situation of |H〉. Therefore, Bob can choose to switch different polarized states to the detector $D_{2}$ by the control of the switch time.

(3) At last, Alice and Bob announce which detector clicks. If only $D_{1}$ detects a photon with the correct polarization, they establish a key bit, otherwise, the result will be used to detect eavesdropping. In fact, if Alice’s and Bob’s bits are identical, the pulse in path *b* will be absorbed by $D_{2}$, and the pulse in path *a* will be divided into two parts towards $D_{0}$ and $D_{1}$, respectively. In this situation, the three detectors $D_{0}$, $D_{1}$ and $D_{2}$ will click with the probabilities $R^{2}$, RT and *T*, respectively. If Alice’s and Bob’s bits are different, the pulse in path *b* will be reflected back to BS. The faraday mirror (FM) alters the state of the pulse to the orthogonal state while reflects it, therefore, the pulse will determinately pass the OL once and be reflected by the PBS twice, before or after the reflection of FM. The two paths *a* and *b* are set with the same length, so the two pulses will complete the interference at BS, with the same polarization state and a phase difference of π. In this situation, $D_{0}$ always clicks but $D_{1}$ never. Therefore, Alice and Bob would share an identical bit when only $D_{1}$ clicks. In ideal cases, the shared bit is secure since no photons have passed through the public channel if $D_{1}$ clicks alone.

Generally, to achieve the highest key rate, *R* and *T* are set to be 1/2 and 1/2. Considering the situation of 50:50 BS, there is an alternative version (see Figure 2) of the above protocol.

In this alternative version proposed by Brida et al. [33], Bob uses a half wave plate (HWP) and a PBS to accomplish the same task with that in the original protocol. The effect of the half wave plate can be described as follows,
(3)$$U\left(\alpha \right)=i\left[\begin{array}{cc} \cos2\alpha & \sin2\alpha \\ \sin2\alpha & -\cos2\alpha \end{array}\right],$$
where α is the angle between the incident and the fast axis. The two faraday mirrors are replaced by two mirrors. In Step (1), by adjusting the angle of HWP_*A*_, Alice randomly rotates the state of the single-photon pulse to |H〉 or |V〉. In Step (2), Bob randomly performs U(0) or U(π/4) to the coming pulse by adjusting the angle of HWP_*B*_ to be 0 or π/4, where
(4)$$\begin{array}{ccc} U(0) & = & i|H\rangle \langle H|-i|V\rangle \langle V|, \end{array}$$
(5)$$\begin{array}{ccc} U(\pi /4) & = & i|H\rangle \langle V|+i|V\rangle \langle H|. \end{array}$$

For convenience of reading, we list another two operations, which will be used later,
(6)$$\begin{array}{ccc} U(\pi /8) & = & \frac{i}{\sqrt{2}}(|H\rangle \langle H|+\left|H\rangle \langle V\right|+\left|V\rangle \langle H\right|-|V\rangle \langle V|), \end{array}$$
(7)$$\begin{array}{ccc} U(3\pi /8) & = & \frac{i}{\sqrt{2}}(-|H\rangle \langle H|+\left|H\rangle \langle V\right|+\left|V\rangle \langle H\right|+|V\rangle \langle V|). \end{array}$$

When the pulse passes BS the first time, the state becomes one of the following states,
(8)$$\begin{array}{c} \rho_{BS}\left(0\right)=-\frac{i}{\sqrt{2}}{|0\rangle }_{a}{|V\rangle }_{b}+\frac{1}{\sqrt{2}}{|V\rangle }_{a}{|0\rangle }_{b}, \end{array}$$
(9)$$\begin{array}{c} \rho_{BS}\left(\pi /4\right)=\frac{i}{\sqrt{2}}{|0\rangle }_{a}{|H\rangle }_{b}-\frac{1}{\sqrt{2}}{|H\rangle }_{a}{|0\rangle }_{b}, \end{array}$$
which are exactly the same with Equations (Equation 1) and (2), ignoring the global phase −i or i introduced by HWP_*A*_. Here, we assume that S always emits a pulse in state |V〉. When Alice’s and Bob’s choices are 0 and π/4, respectively, the pulse in path *b* has been reflected back to BS at Alice’s side, the state of the photon after the pulse passes BS the second time will become
(10)$$\begin{array}{ccc} \rho_{BS}^{\prime }\left(0,\pi /4\right) & = & -\frac{1}{\sqrt{2}}{(\frac{1}{\sqrt{2}}|V\rangle }_{0}{|0\rangle }_{1}{|0\rangle }_{2}+\frac{i}{\sqrt{2}}{|0\rangle }_{0}{|V\rangle }_{1}{|0\rangle }_{2})+\frac{i}{\sqrt{2}}{(\frac{i}{\sqrt{2}}|V\rangle }_{0}{|0\rangle }_{1}{|0\rangle }_{2}+\frac{1}{\sqrt{2}}{|0\rangle }_{0}{|V\rangle }_{1}{|0\rangle }_{2})\\ & = & -{|V\rangle }_{0}{|0\rangle }_{1}{|0\rangle }_{2}, \end{array}$$
or similarly when Alice’s and Bob’s choices are π/4 and 0, the state would be
(11)$$\rho_{BS}^{\prime }\left(\pi /4,0\right)={|H\rangle }_{0}{|0\rangle }_{1}{|0\rangle }_{2},$$
where the subscripts represent the path leading to the corresponding detectors. Thus, the same as the original protocol, $D_{0}$ always clicks in this situation. Correspondingly, the key should be generated when $D_{1}$ clicks.

## 3. QIA in the Counterfactual QKD System

In this section, we propose a QIA protocol, where two participants, utilizing a pre-shared classical authentication key, can verify each other’s identity through the counterfactual QKD system. The communication system we adopt here is the alternative version, which is more convenient to introduce a conjugate basis to complete the task of identity authentication. Here, we make a minor modification that the half wave plate on Alice’s side, i.e., HPB_*A*_ is set at the right side of BS (see Figure 3).

Thus, the states of the photons back to Alice’s side should always be the same with their original states, and the key bits should be generated from the signals where $D_{0}$ clicks alone.

### 3.1. The QIA Protocol Based on the Counterfactual QKD

For the sake of description of the proposed QIA protocol, we first expound some basic concepts about the protocol and the devices in Figure 3. Before the protocol, two participants are required to pre-share a sequence of authentication keys $\left\{K_{1},K_{2},\ldots ,K_{l}\right\}$. Each of the above keys has *m*+*n* bits, where the first *m* bits would be used for Alice to verify Bob’s identity, and the last *n* bits are for Bob to verify Alice’s identity. Alice and Bob also record the statuses of their keys, originally “valid”.

The single-photon source S in Figure 3 is supposed to always emit a pulse in state |V〉, and Alice can choose to keep its state or flip it to |H〉 utilizing HWP_*A*_. Ignoring the global phases, if Alice adjusts $\alpha_{A}$, the angle of HWP_*A*_, to 0, the state of the pulse remains |V〉, and if Alice adjusts $\alpha_{A}$ to π/4, the state changes to |H〉. Bob also randomly chooses to flip the state of the coming pulse or not, utilizing HWP_*B*_. The above processes are just the alternative version of the counterfactual QKD protocol.

To complete the task of identity authentication, the participants use the authentication key as the control bits in the manner that, if the *i*th bit of the authentication key is 1, Alice and Bob both rotate an additional angle of π/8 to their half wave plates, otherwise, they do nothing additionally. Thus, only the legal participants who have the authentication key can perform complete the QIA protocol legally. The concrete processes of the proposed QIA protocol are as follows.

**1. Key status exchange**. Alice and Bob exchange the status of their pre-shared authentication keys and choose the one with the smallest subscript among those keys which are “valid” on both Alice’s and Bob’s sides. We denote the bits of this key *K* as
(12)$$\{b_{1},b_{2},\ldots ,b_{m},a_{1},a_{2},\ldots ,a_{n}\}.$$

**2. Authentication of Bob’s identity**. The first *m* pulses are used to authenticate Bob’s identity in the manner that Alice chooses her bit randomly and Bob always chooses bit 0, and both of the above choices are under control of the first *m* bits of *K*.
2.1Alice generates a random string $R_{A}$ with *m* bits
(13)$$\{r_{1},r_{2},\ldots ,r_{m}\}.$$2.2For the *i*th pulse Alice emits into the system, she sets the angle of HWP_*A*_ as
(14)$$\frac{\pi }{4}\times r_{i}+\frac{\pi }{8}\times b_{i}.$$2.3For the *i*th coming pulse, Bob sets the angle of HWP_*B*_ as
(15)$$\frac{\pi }{8}\times b_{i}.$$2.4Alice checks the results of $D_{0}$ and $D_{1}$. If $D_{1}$ clicks with the probability of 100% for the pulses where $r_{i}$ = 1, and for those $r_{i}$ = 0, $D_{0}$ and $D_{1}$ click with the probability about 25% and 25%, respectively, Alice believes Bob’s identity and they go on to Step 3, otherwise, Alice skips to the last step.

**3. Authentication of Alice’s identity**. In this step, Bob checks Alice’s identity with the help of the last *n* bits of *K*.
3.1Bob generates a random string $R_{A}$ with *m* bits
(16)$$\{s_{1},s_{2},\ldots ,s_{m}\}.$$3.2For the (m+j)th pulse, Alice sets the angle of HWP_*A*_ as
(17)$$\frac{\pi }{8}\times a_{i}.$$3.3For the (m+j)th coming pulse, Bob sets the angle of HWP_*B*_ as
(18)$$\frac{\pi }{4}\times s_{i}+\frac{\pi }{8}\times a_{i}.$$3.4Bob checks results of $D_{2}$. If $D_{2}$ never clicks when $s_{i}$ = 0 and clicks with the probability of 50% for both the two cases that $\{s_{i}$ = 1,$a_{i}$ = 0} and $\{s_{i}$ = 1,$a_{i}$ = 1}, Bob believes Alice’s identity.

**4. Key status update**. Alice and Bob update the statuses of *K* as “invalid”.

### 3.2. Correctness of the Proposed QIA Protocol

For the legal Alice and Bob, they can verify each other’s identity following the above processes. The unitary operations of the HWPs in different cases are shown in Equations (Equation 4)–(Equation 6). In the processes of Step 2, there are four cases about Alice’s and Bob’s choices of $\left\{\alpha_{A},\alpha_{B}\right\}$: {0,0}, {π/4,0}, {π/8,π/8}, and {3π/8,π/8}. For the situation of {0,0}, the state of the whole pulse when it first passes BS and HWP_*A*_ is
(19)$$\rho_{HA}\left(0\right)=-\frac{i}{\sqrt{2}}{|0\rangle }_{a}{|V\rangle }_{b}+\frac{i}{\sqrt{2}}{|V\rangle }_{a}{|0\rangle }_{b}.$$

The final state, i.e., the state of the polarization and the position of the photon after (part of) it passes BS the second time, is
(20)$$\begin{array}{ccc} \rho_{BS}^{\prime }\left(0,0\right) & = & -\frac{1}{\sqrt{2}}{(\frac{i}{\sqrt{2}}|V\rangle }_{0}{|0\rangle }_{1}{|0\rangle }_{2}+\frac{1}{\sqrt{2}}{|0\rangle }_{0}{|V\rangle }_{1}{|0\rangle }_{2})-\frac{i}{\sqrt{2}}{|0\rangle }_{0}{|0\rangle }_{1}{|V\rangle }_{2} \end{array}$$
(21)$$\begin{array}{cc} = & -\frac{i}{2}{|V\rangle }_{0}{|0\rangle }_{1}{|0\rangle }_{2}-\frac{1}{2}{|0\rangle }_{0}{|V\rangle }_{1}{|0\rangle }_{2}-\frac{i}{\sqrt{2}}{|0\rangle }_{0}{|0\rangle }_{1}{|V\rangle }_{2}. \end{array}$$

Here, we use $\rho_{HA}\left(\alpha \right)$ to denote the state of the pulse when it first passes BS and HWP_*A*_ in the situation that $\alpha_{A}$=α, $\rho_{PBS}\left(\alpha_{1},\alpha_{2}\right)$ to denote the state when the pulse first passes PBS in the situation that $\alpha_{A}$=$\alpha_{1}$ and $\alpha_{B}$=$\alpha_{2}$, and $\rho_{BS}^{\prime }\left(\alpha_{1},\alpha_{2}\right)$ to denote the state when the pulse passes BS the second time. For the situation of {π/8,π/8},
(22)$$\begin{array}{ccc} \rho_{HA}\left(\pi /8\right) & = & \frac{i}{\sqrt{2}}{|0\rangle }_{a}{|-\rangle }_{b}+\frac{i}{\sqrt{2}}{|V\rangle }_{a}{|0\rangle }_{b}, \end{array}$$
where
(23)$$\begin{array}{c} |-\rangle =\frac{1}{\sqrt{2}}(|H\rangle -|V\rangle ). \end{array}$$
and
(24)$$\begin{array}{c} \rho_{PBS}\left(\pi /8,\pi /8\right)=-\frac{1}{\sqrt{2}}{|0\rangle }_{a}{|V\rangle }_{b}+\frac{i}{\sqrt{2}}{|V\rangle }_{a}{|0\rangle }_{b}. \end{array}$$

Thus, the final state of this case would be
(25)$$\begin{array}{c} \rho_{BS}^{\prime }\left(\pi /8,\pi /8\right)=-\frac{i}{2}{|V\rangle }_{0}{|0\rangle }_{1}{|0\rangle }_{2}-\frac{1}{2}{|0\rangle }_{0}{|V\rangle }_{1}{|0\rangle }_{2}-\frac{i}{\sqrt{2}}{|0\rangle }_{0}{|0\rangle }_{1}{|V\rangle }_{2}. \end{array}$$

For both above cases of Equations (Equation 20) and (Equation 25), $D_{0}$, $D_{1}$ and $D_{2}$ would click with the probabilities of 25%, 25% and 50%, respectively. Similarly, we can calculate that
(26)$$\begin{array}{c} \rho_{BS}^{\prime }\left(\pi /4,0\right)=\rho_{BS}^{\prime }\left(3\pi /8,\pi /8\right)=-{|0\rangle }_{0}{|V\rangle }_{1}{|0\rangle }_{2}. \end{array}$$

$D_{1}$ always clicks in these two cases. The calculations on the four cases coincide with the judgements at the end of Step 2.

The four possible final states in the processes of Step 3 are $\rho_{BS}^{\prime }\left(0,0\right)$, $\rho_{BS}^{\prime }\left(0,\pi /4\right)$, $\rho_{BS}^{\prime }\left(\pi /8,\pi /8\right)$, and $\rho_{BS}^{\prime }\left(\pi /8,3\pi /8\right)$. Here, $\rho_{BS}^{\prime }\left(0,0\right)$ and $\rho_{BS}^{\prime }\left(\pi /8,\pi /8\right)$ are considered in Equations (Equation 20) and (Equation 25), and
(27)$$\begin{array}{c} \rho_{BS}^{\prime }\left(0,\pi /4\right)=\rho_{BS}^{\prime }\left(\pi /8,3\pi /8\right)=-{|0\rangle }_{0}{|V\rangle }_{1}{|0\rangle }_{2}. \end{array}$$

The calculations on these four final states in Step 3 coincide with the judgements in Step 3, too. Therefore, for the legal participants who have the authentication key *K*, they can always authenticate each other’s identities correctly.

### 3.3. The Security Analysis for No-Error Cases

In fact, the security of Bob’s identity is protected by the first part of *K*, i.e., $\left\{b_{1},b_{2},\ldots ,b_{m}\right\}$. The operations Bob’s operations in the first part is U(0) if $b_{i}$= 0 and U(π/8) if $b_{i}$ = 1. According to the theorems on operation discrimination, the above two operations cannot be discriminated with no error probability (see Appendix A for details). If the adversary, who is forging Bob’s identity to communicate with Alice, performs an error operation on the received pulse, Alice would get a wrong measurement result. Therefore, the adversary cannot always gives Alice a correct response to pass Alice’s tests. Correspondingly, the security of Alice’s identity is protected by the second part of *K*, i.e., $\left\{a_{1},a_{2},\ldots ,a_{n}\right\}$. Since the second part of the protocol only executed when the communicating peer passes Alice’s tests, the adversary cannot get any information about the string $\left\{a_{1},a_{2},\ldots ,a_{n}\right\}$ from Alice’s side. Therefore, the adversary has to face Bob’s tests without any information on the authentication key. Considering that Bob chooses his angles from {0,π/8π/4,3π/8} randomly, the measurement results would be random if he is communicating with the adversary who has no information about $\left\{a_{1},a_{2},\ldots ,a_{n}\right\}$. That is, the adversary cannot pass Bob’s tests without introducing any error. Above all, we can get a conclusion that the adversary cannot forge either Alice’s identity or Bob’s identity in the no-error cases.

For the more general cases, the security analysis of the proposed protocol are described in Appendix A. Specifically, for each signal, we calculate a relaxed lower bound of the minimum error probability that the adversary has to introduce in Alice’s test while forging Bob,
(28)$$P_{b}=\frac{1}{2}\left(1-\frac{\sqrt{5-2\sqrt{2}}}{16\sqrt{2}-8}\right)>2.8\%,$$
and a relaxed lower bound of the minimum error probability that the adversary has to introduce in Bob’s test while forging Alice,
(29)$$P_{a}>6.5\%.$$


We believe that the tight bounds would be much larger, since we have made many relaxations during the derivation procedure to simplify the difficulty.

## 4. Authenticated Counterfactual QKD Protocol

In this section, we propose an authenticated counterfactual QKD protocol utilizing the proposed QIA protocol. The basic idea is mixing the process of the QIA protocol into the QKD protocol according to the random data generated in the QKD protocol, which can be recorded identically for the two participants without any communication. In the following authenticated QKD protocol, the length of original authentication key is independent with the length of the new generated key. Suppose the length of the key that the participants expect to generate is *m*, and the length of the authentication key $K_{A}$ which meets the requirement of security is *n*. Then, the mixing parameter of the authenticated counterfactual QKD protocol is
(30)$$r=\lfloor \frac{4m}{n}\rfloor .$$

With the definition of *r*, the main processes of the authenticated QKD protocol can be briefly described as follows: once Bob’s detector has clicked *r* times, Alice and Bob insert one round of the QIA process presented in last section. Specifically, utilizing the devices and circuit in Figure 3, the participants can implement the authenticated counterfactual QKD protocol as follows. For convenience of the following description, we use $p_{i}$ to denote the probability that the *i*th signal is used for the process of QIA.

***a*****. Set-up**. For the main processes described above, $p_{i}$ is convergent when *i* gets larger, however it is much smaller than the convergence value for small *i*s. For example, $p_{i}$ = 0 when *i* ≤ *r*. If the adversary only attacks these signals with smaller $p_{i}$, he is more likely to pass the participant’s test. Therefore, before the formal steps of the protocol, Alice and Bob should equalize $p_{i}$ for different *i*s. $l_{r}$ pulses would be used in this stage, where
(31)$$l_{r}=2\left\lceil \log\left(4r+1\right)\right\rceil .$$

$a_{1}$Alice emits $l_{r}$ single-photon pulses to the system one by one. For each pulse, Alice (Bob) randomly choose the angle of HWP_*A*_ (HWP_*B*_) to be one of {0, π/8, π/4, 3π/8}.$a_{2}$If the photon goes to Bob’s detector, i.e., $D_{2}$ clicks and $D_{0}$ and $D_{1}$ do not, they record a classical bit 1. If the photon goes back to Alice, i.e., $D_{0}$ or $D_{1}$ clicks and $D_{2}$ does not, they record a classical bit 0.$a_{3}$After all the $l_{r}$ pulses have been detected by the three detectors, Alice and Bob get a $l_{r}$ bit binary number. Then, they use a hash function to uniformly map the above number into the set {0,1,…,4r}, and denote the result as $f_{r}$. Note that, for one single binary bit, the uncertainty is
(32)$$-\frac{1}{4}\log\left(\frac{1}{4}\right)-\frac{3}{4}\log\left(\frac{3}{4}\right)\approx 0.56.$$Alice and Bob produce $l_{r}$ signals here so that the uncertainty of the $l_{r}$ bits is larger than log(4r+1), to make the value of $f_{r}$ totally random.

***b*****. Signal transmission and identity authentication**. Utilizing the random number $f_{r}$ generated in last step, the participants start to distribute a new key while authenticate each other’s identity.
$b_{1}$For the first $f_{r}$ pulses in this step, Alice and Bob perform the QKD process, i.e., they both randomly alter the angles of HWP_*A*_ and HWP_*B*_ to be 0 or π/4 and record the clicking situation of each detector and the state of the photon if the detector has clicked.$b_{2}$The $\left(f_{r}+1\right)$th pulse is the first pulse for identity authentication. As in Steps 2.2 and 2.3 in the above QIA protocol, Alice alters the angle of HWP_*A*_ to be $\pi /4*r_{1}$+$\pi /8*b_{1}$ and Bob alters the angle of HWP_*B*_ to be $\pi /8*b_{1}$, where $b_{1}$ is the first bit of the authentication key and $r_{i}$ is a random bit.$b_{3}$From the $\left(f_{r}+2\right)$th pulse, the participants start to insert the process of QIA into the QKD according to the random data of the clicks of the detectors. Precisely, each time the click times of $D_{2}$ reaches an integral multiple of *r*, they insert one round of the QIA process immediately until the authentication process for Bob’s identity has finished.$b_{4}$Alice checks Bob’s identity according to Step 2.4.$b_{5-8}$If the test for Bob’s identity passes, they continue to transmit the rest QKD signals and authenticate Alice’s identity by repeating the processes from $b_{1}$ to $b_{4}$ but perform the operations in Steps 3.2–3.4 instead of these in Steps 2.2–2.4, respectively.

***c*****. Eavesdropping detection**. Alice and Bob first check the validity of each other’s identity. If the identity authentication passes, they continue to the rest part of the counterfactual QKD protocol to generate a new key and use part of the new key to update the authentication keys. 

In the above protocol, the processes of QIA and QKD are mixed randomly; however, they are performed independently. More specifically, the sequence of the signals for QIA and ones for QKD are random for the adversaries. On the other hand, each signal is either used for QIA or for QKD, but never for both. Because of such independence, the correctness of the above protocol is obliviously established considering the correctness of the counterfactual QKD protocol [27,31] and the counterfactual QIA protocol presented in the last section, as is the security of the process of QIA and the process of QKD. The only new factor which may influence the security of the whole protocol is that the adversary may discriminate the two type of signals, i.e., the signals for QKD and the signals for QIA, and then only attack the signals for QKD. Next, we proved that the adversary cannot discriminate the two type of signals.

Firstly, the two types of signals cannot be discriminated precisely. For a QKD signal, Alice randomly sets her angle as 0 or π/4, and the reduced density matrix for the state in path *b* is
(33)$$\begin{array}{ccc} \rho_{QKD} & = & \frac{1}{2}(\frac{1}{2}(|0\rangle \langle 0|+|V\rangle \langle V|)+\frac{1}{2}(|0\rangle \langle 0|+\left|H\rangle \langle H|)\right)\\ & = & \frac{1}{2}\left|0\rangle \langle 0\right|+\frac{1}{4}I. \end{array}$$

For a QIA signal in the first part, Alice’s operation set is {0,π/8,π/4,3π/8}, and the reduced density matrix for the state in path *b* is
(34)$$\begin{array}{ccc} \rho_{QIA_{B}} & = & \frac{1}{4}(\frac{1}{2}(|0\rangle \langle 0|+|V\rangle \langle V|)+\frac{1}{2}(|0\rangle \langle 0|+|H\rangle \langle H|)\\ & & +\frac{1}{2}(|0\rangle \langle 0|+|+\rangle \langle +|)+\frac{1}{2}(|0\rangle \langle 0|+|-\rangle \langle -|))\\ & = & \frac{1}{2}\left|0\rangle \langle 0\right|+\frac{1}{4}I, \end{array}$$
which is the same as $\rho_{QKD}$. For the second part of QIA, Alice’s operation set is {0,π/8}, and the reduced density matrix for the state in path *b* is
(35)$$\begin{array}{ccc} \rho_{QKA_{A}} & = & \frac{1}{2}(\frac{1}{2}(|0\rangle \langle 0|+|V\rangle \langle V|)+\frac{1}{2}(|0\rangle \langle 0|+|-\rangle \langle -|))\\ & = & \frac{1}{2}\left|0\rangle \langle 0\right|+\frac{1}{4}(|V\rangle \langle V|+|-\rangle \langle -|) \end{array}$$

The minimum error probability to discriminating $\rho_{QKD}$ and $\rho_{QKA_{A}}$ is
(36)$$\frac{1}{2}-\frac{\sqrt{2}}{16}\approx 0.41,$$
which is close to 1/2, the probability of random guess. Furthermore, the discrimination operation will inevitably disturb the pulse in path *b* and introduce errors in the authentication process or the detection mode of QKD.

Secondly, we analyze the probability of being a QIA pulse for each signal. The expectation of the above probability can be deduced by calculating the average interval of two QIA signals, which is
(37)$$\begin{array}{ccc} D & = & \sum_{j=r}^{\infty }jpC_{j-1}^{r-1}p^{r-1}{\left(1-p\right)}^{j-r}. \end{array}$$
(38)$$\begin{array}{cc} = & p^{r}\sum_{k=0}^{\infty }\left(r+k\right)C_{r+k-1}^{k}{\left(1-p\right)}^{k}. \end{array}$$

Then, both sides of the above equation are multiplied by (1−p),
(39)$$\begin{array}{c} \left(1-p\right)D=p^{r}\sum_{k=0}^{\infty }\left(r+k-1\right)C_{r+k-2}^{k-1}{\left(1-p\right)}^{k}. \end{array}$$

By subtracting the two equations, we have
(40)$$\begin{array}{ccc} pD & = & p^{r}\sum_{k=1}^{\infty }\left(r+k-1\right)\left(C_{r+k-1}^{k}-C_{r+k-2}^{k-1}\right){\left(1-p\right)}^{k}+rp^{r}++p^{r}\sum_{k=1}^{\infty }C_{r+k-2}^{k-1}{\left(1-p\right)}^{k}\\ & = & rp^{r}+p^{r}\sum_{k=1}^{\infty }{\left(1-p\right)}^{k}\left(\left(r+k\right)C_{r+k-2}^{k}+C_{r+k-2}^{k-1}\right)\\ & = & rp^{r}+rp^{r}\sum_{k=1}^{\infty }C_{r+k-1}^{k}{\left(1-p\right)}^{k}. \end{array}$$

Suppose
(41)$$D_{i}=\sum_{k=i}^{\infty }C_{r+k-i}^{k}{\left(1-p\right)}^{k}.$$

Then,
(42)$$\begin{array}{ccc} \left(1-p\right)D_{i} & = & \sum_{k=i}^{\infty }C_{r+k-i}^{k}{\left(1-p\right)}^{k+1}=\sum_{k=i+1}^{\infty }C_{r+k-(i+1)}^{k-1}{\left(1-p\right)}^{k}. \end{array}$$

We can get
(43)$$\begin{array}{c} pD_{i}={\left(1-p\right)}^{i}C_{r}^{i}+D_{i+1}. \end{array}$$

So that,
(44)$$\begin{array}{ccc} D_{1} & = & \sum_{k=1}^{\infty }C_{r+k-1}^{k}{\left(1-p\right)}^{k}\\ & = & \sum_{j=1}^{r}C_{r}^{j}{\left(1-\frac{1}{p}\right)}^{j}+\frac{1}{p^{r}}\sum_{k=r}^{\infty }C_{k-1}^{k}{\left(1-p\right)}^{k}\\ & = & \frac{1}{p^{r}}-1. \end{array}$$

Substituting Equation (Equation 44) into Equation (Equation 40), we have
(45)$$D=\frac{r}{p},$$
where *p* = 1/4. This implies that every *D*+1 signals contain one QIA signal on average. Therefore, the average probability for a pulse to be a QIA signal is
(46)$$\bar{p}=\frac{1}{4r+1},$$

However, it is difficult to propose a strategy where the above probability is totally identical for each signal. As for the proposed protocol, the probability of the *l*th signal to be a QIA one is
(47)$$\begin{array}{c} P_{r}\left(l\right)=\bar{p}\sum_{k=1}4r+1\sum_{j=1}^{\left\lfloor \frac{l}{r+1}\right\rfloor }C_{l-1-i-k}^{rj-1}{\left(\frac{1}{4}\right)}^{rj}{\left(\frac{3}{4}\right)}^{l-rj-j-k}. \end{array}$$

The the graphs of function $p_{r}\left(l\right)$ for different values of *r* are given in Figure 4.

We can see that the probability $p_{l}$ tends to be stable when *l* is larger than 8r. The adversary cannot effectively reduce the error rate introduced by his attack utilizing the probability distribution of the type of the signals. Therefore, if the adversary wants to attack a proportion of the QKD signals, she will have to disturb a similar proportion of the QIA signals, which will cause a failure result in the QIA part.

## 5. Conclusions

In this paper, we first propose a quantum identity authentication protocol that can be realized in the counterfactual quantum communication system. Then, we propose an authenticated counterfactual QKD protocol by mixing the processes of the proposed QIA protocol and the counterfactual QKD protocol in [33]. In this authenticated counterfactual QKD protocol, the two independent processes of QKD and QIA mixed randomly for any third party except the two participants, therefore, the adversaries cannot discriminate between a QIA signal and a QKD signal. Any attempts to perform a man-in-the-middle attack to the process of QKD will disturb the signal in the QIA process and cause a failure result in the identity authentication. Since the two processes are independent, the length of the authentication key is only related to the expected confidence degree for the participants’ identities, and is not concerned with length of the newly generated key in QKD. Therefore, the key expansion in our protocol can be extremely high in theory. The problem is that the proposed protocol can only be performed in noiseless channels since any channel loss or dark count would mess up the whole process of the protocol. Once a channel loss or dark count happens, Alice and Bob cannot synchronize the random data to control the signal type. Despite this, we think the idea of identity authentication in this paper is promising in theory and might inspire practical QIA protocols and authenticated QKD protocols designed in similar ways. The theory of high key-expand-ability QIA protocols in noisy channel will also be our future work.

## Figures and Tables

**Figure 1 entropy-21-00518-f001:**
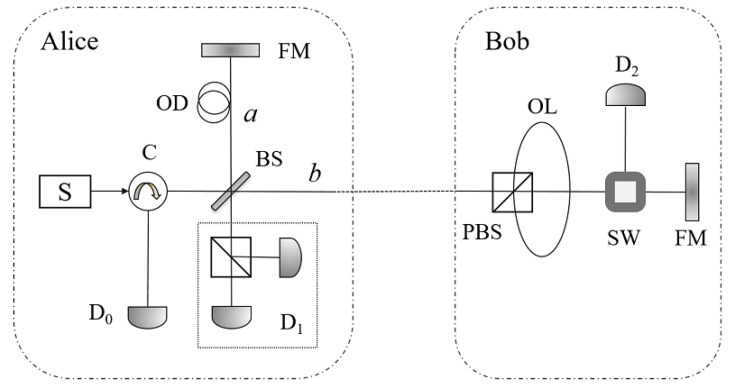
The schematic of the counterfactual QKD [27]. Here, C is the optical circulator; OD is the optical delay to make the two paths *a* and *b* be the same; OL is the optical loop and SW is the optical switch, which help Bob choose the pulse in specific polarization to the detector D_2_; FM is the faraday mirror, which reflects the pulse while turns the state of the pulse to the orthogonal polarization; and D_1_ can discriminate the polarizations of the pulse.

**Figure 2 entropy-21-00518-f002:**
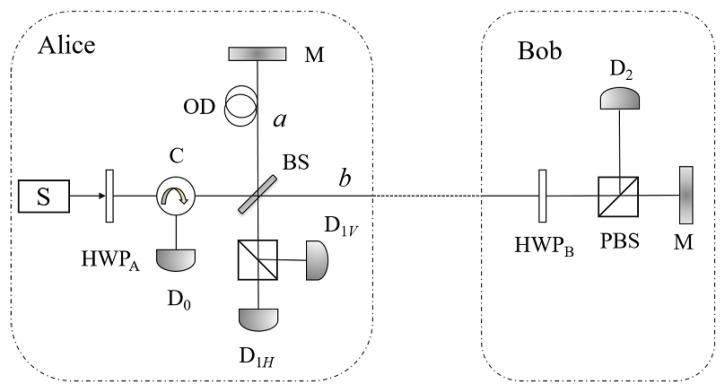
The schematic of the alternative version of the counterfactual QKD [33]. The alternative version uses two half wave plates HWP_*A*_ and HWP_*B*_, instead of the OL and SW, to implement the random choices of the participants. Another difference is that the alternative version uses mirrors (M) instead of faraday mirrors in the original one.

**Figure 3 entropy-21-00518-f003:**
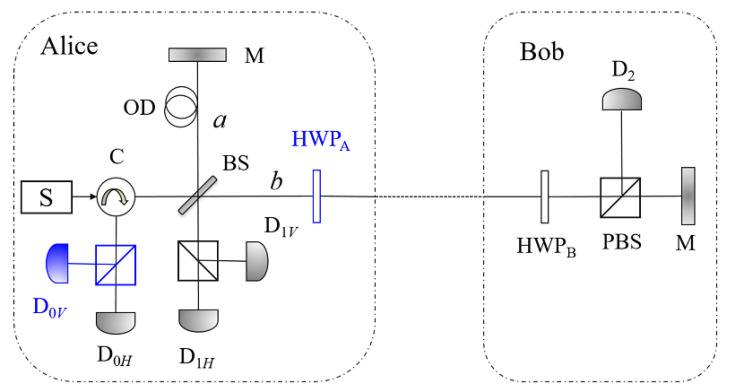
The schematic of the proposed QIA protocol and the authenticated QKD protocol. HWP_*A*_ is set at right side of BS in our protocol instead of left in the alternative version. Since the polarizations of the pulses detected by D_0_ is also used to detect the adversary in our protocol, we add a PBS and an additional detector in the D_0_.

**Figure 4 entropy-21-00518-f004:**
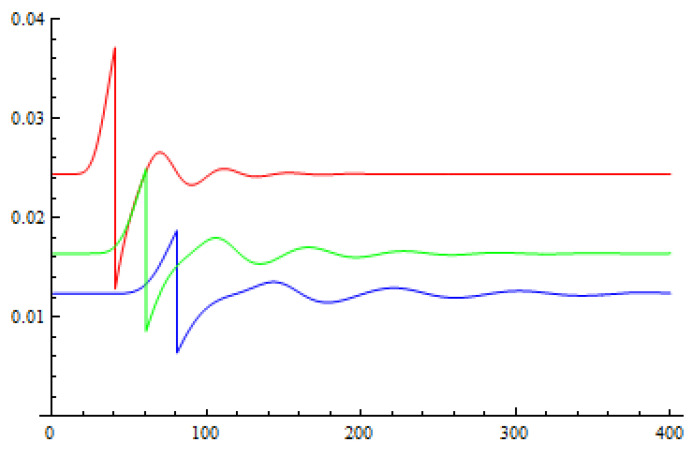
The graphs of function $p_{r}\left(l\right)$ when *r* = 10 (the red line), 15 (the green line), and 20 (the blue ine).

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
