# Peer review of "Quantum Identity Authentication in the Counterfactual Quantum Key Distribution Protocol"

_entropy, 2019, doi:10.3390/e21050518_

Round 1

Reviewer 1 Report

My comments on this work are:

1) This work has a good exposition of the technique and protocol developed.

2) The method presented in the present work is an improvement of previous works, presenting a proposal of a quantum identity authentication protocol which can be realized in the counterfactual quantum communication systems whith noiseless channels

3) The proposal is original and could be implemented, with some small modifications, in QIA protocols that have authenticated QKD protocols.

4) Theoretically, the counterfactual QKD scheme presented in Figures 1 and 2, and the proposal of a QIA protocol and the authenticated QKD protocol, are correct and well analyzed and explained in the presented calculations.

For the reasons stated above, I recommend the acceptance for publication of this paper in Entropy, after the following minor corrections are done:

In the line 69 it is not clear why the authors say that "... to the D3 detector according to the above bit.", Since in Figure 1 there is not D3 detector. It seems to be a writing error.

Author Response

Thank you for your kindness comments 1-4.

And we are sorry about the  low-level mistakes in our manuscript.

We have corrected the mistake you mentioned, and it should be D2 rather than D3.

And we also checked our manuscript  carefully for other description mistakes.

Reviewer 2 Report

In this manuscript, the authors give a method for authenticating legitimate users in a quantum key distribution (QKD) system. They begin with a previously published setup for counterfactual QKD and make changes to the procedure in order to foil eavesdroppers while allowing the users to verify each other's pre-distributed classical authentication keys. In counterfactual QKD, qubits are generated by interference between multiple potential paths, and no qubits can be said to travel between Alice and Bob. After reviewing counterfactual QKD, the authors adapt the setup (with a few added components) and the protocol to carry out a form of counterfactual user authentication. They describe the protocol in detail, and show how it can be incorporated with a key distribution scheme to form a full user-authenticated counterfactual QKD system. They then analyze security. In particular, they show (under fairly conservative assumptions) that the minimum error introduced by the eavesdropper into the Alice's and Bob's tests of each other's identities is 2.8% and 6.5% The probability that Alice will detect Eve's presence grows exponentially with the length m of Bob's authentication key: it is at least 1-.97^m. Under slightly more relaxed assumptions, the eavesdropper-induced error rate should go up further. The paper is clear and well-written, and I see no problems with the calculations or the procedure. Although it builds on a previously published protocol, it repurposes the protocol for a new purpose, introducing a degree of novelty. The paper should be of reasonable usefulness and interest to the quantum information community, and I am willing to recommend it to be published essentially in the current form. The only minor suggestion I would recommend (although it is not essential) concerns the setup in figure 1, which comes from reference 27; although 27 is referenced in description in the main text, it should also be cited in the caption so it is clear where the figure came from. Similarly for figure 2 and reference 33.

Author Response

Thank you for your kindness comments.

We have modified the manuscript followed your comments.

In this new version, figure 1 and figure 2 have both cited the original literatures where they came from.